# Food insecurity and mental health among migrants and refugees in high-income countries: Systematic review and meta-analyses

Resom Berhe [1,2]*, Amit Arora [2,3,4,5,6,7], Kanchana Ekanayake[8], Kingsley E. Agho[2,3,9,10]

1 Department of Health Education and Behavioural Sciences, University of Gondar, College of Medicine and Health Science, Institute of Public Health, Gondar, Ethiopia, 2 School of Medicine, Faculty of Health, Western Sydney University, Sydney, New South Wales, Australia, 3 Translational Health Research Institute, Western Sydney University, Sydney, New South Wales, Australia, 4 Discipline of Child and Adolescent Health, Sydney Medical School, Faculty of Medicine and Health, The University of Sydney, Westmead, New South Wales, Australia, 5 Oral Health Services, Sydney Local Health District and Sydney Dental Hospital, NSW Health, Surry Hills, Sydney, New South Wales, Australia, 6 Health Equity across Lifespan (HEAL) Research Laboratory, Campbelltown, New South Wales, Australia, 7 ARCED Foundation, Dhaka, Bangladesh, 8 University of Sydney Library, Camperdown, New South Wales, Australia, 9 Faculty of Health Sciences, University of Johannesburg, Johannesburg, South Africa, 10 Global Maternal and Child Health Research collaboration (GloMACH), Western Sydney University, Campbelltown, New South Wales, Australia

* resom.berhe86@gmail.com

## Abstract

### Background

Food insecurity (FI) is a recognised determinant of mental health, yet its specific impact on depression, anxiety, and stress among migrants and refugees in high-income countries (HICs) remains underexplored. This systematic review and meta-analyses examine the association between FI and mental health outcomes in these vulnerable populations.

### Method

A systematic search of five databases (Medline, Scopus, Web of Science, PsycINFO, and Embase) was conducted for observational studies published between January 1, 2008, and December 16, 2025. Inclusion required studies to report on food insecurity and mental health using standard, validated measures. Three independent reviewers screened the studies, extracted the data, and assessed the methodological quality using the Joanna Briggs Institute checklist. Most included studies were cross-sectional. The association between food insecurity and mental health outcomes (depression, anxiety, and stress) was estimated using random-effects models due to substantial heterogeneity ($I^2 > 50\%$). Funnel plots and Begg's test were employed to assess potential publication bias.

**Data availability statement:** All relevant data are within the paper and its Supporting information files.

**Funding:** The author(s) received no specific funding for this work.

**Competing interests:** The authors have declared that no competing interests exist.

## Result

We identified 17 eligible studies, totalling more than 4.3 million participants. Overall, FI was significantly associated with higher odds of depression (aOR = 1.44; 95% CI: 1.32–1.57), anxiety (aOR = 2.43; 95% CI: 1.82–3.23), and stress (aOR = 5.9; 95% CI: 2.95–11.94) among migrants and refugees in HICs. Other significant factors were women (aOR = 1.44; 95% CI: 1.32–1.57), Unmarried individuals (aOR = 1.7; 95% CI: 1.49–1.83), and those with poor self-rated health (aOR = 3.6; 95% CI: 2.97–4.36). Moreover, higher household income was significantly associated with lower odds of depression compared to their counterparts (aOR = 0.90; 95% CI: 0.84–0.98; P = 0.007).

## Conclusion

Food insecurity was consistently associated with higher odds of depression, anxiety, and stress among migrants and refugees in high-income countries. These findings support the implementation of enhanced screening and integrated social and mental health responses that address food access and psychological well-being.

## Trial registration

PROSPERO CRD42024525690

## 1. Introduction

Mental health, including depression, anxiety, and stress, is a primary global public health concern [1], with profound social and economic implications. According to recent estimates from the World Health Organization, in 2019, approximately 280 million people worldwide suffered from depression, and about 301 million experienced an anxiety disorder [2]. The COVID-19 pandemic further exacerbated this burden, with global prevalence of depression and anxiety increasing by over 25% in 2020 [2]. Depression alone is associated with nearly a twofold increase in the risk of mortality compared to the general population [2,3], underscoring the need to identify and address factors that heighten mental health vulnerability in marginalized groups. For example, migrants and refugees often face socioeconomic hardship, which significantly increases the risk of mental health disorders. Among the challenges these populations face, food insecurity is a critical yet frequently overlooked determinant of mental well-being. Addressing food insecurity is, therefore, essential to improving mental health outcomes in these populations.

However, despite its evident importance, food insecurity (FI) remains an underexplored factor in mental health research on migrants and refugees. Long described as the "paradox of hunger amid plenty," FI continues to rise globally, affecting over 345 million people in 2023, more than double the figure recorded in 2020 [4]. Although typically associated with low-income regions, FI is increasingly prevalent among migrants and refugees in high-income countries (HICs). In these settings, language

barriers, unstable employment, unfamiliar food systems, and limited social support heighten the risk of FI [5]. As of the end of 2023, over 117 million people were forcibly displaced worldwide, many of whom were resettled in HICs. In these host countries, FI rates among migrants and refugees are often two to three times higher than those of native-born populations [6,7]. FI's causes and consequences are multifaceted, encompassing nutritional, psychological, and social dimensions. Reflecting this multidimensional nature, FI has been linked to a range of adverse health outcomes beyond hunger and malnutrition.

Building on this understanding, a growing body of evidence highlights the broader health implications of FI. FI has been linked to various adverse physical health outcomes, including obesity, anemia, diabetes, hypertension, and asthma [8–11]. More recently, FI has also been recognized as a potent contributor to psychological distress, exacerbating symptoms of anxiety, depression, and stress [12–16]. In addition to hunger itself, the mental toll of FI includes fear, uncertainty, restricted autonomy in food choices, and the stigma of relying on assistance programs [17,18]. For migrants and refugees, who often face precarious legal status, discrimination, and cultural displacement, the stressors associated with FI may be even more pronounced [19–21].

Despite growing recognition of the links between FI and mental health, most existing research has focused on general or nationally representative samples, which often under-represent immigrant experiences. A few reviews have examined specific subgroups. For instance, Maynard et al. (2018) [22] reviewed FI and mental health among women and found a consistent link with postpartum depression. Similarly, Arenas et al. (2019) [23] examined FI and mental well-being in the general population, reinforcing the evidence that FI correlates with psychological distress. However, none of these has focused explicitly on migrant or refugee populations in high-income settings, who face unique circumstances. Migrants and refugees contend not only with the typical stressors of FI but also with additional burdens such as acculturation challenges, unstable legal status, and histories of trauma, which might amplify FI's psychological impact. Moreover, the distribution and determinants of FI in migrant communities might differ from those in the general population. For example, certain cultural factors or migration policies may influence FI in these groups, meaning that general population data may not be fully applicable. To date, no comprehensive review has quantified the prevalence of FI among migrants in HICs, nor systematically analyzed the strength of its association with mental health outcomes in these groups. This represents a significant knowledge gap. Without such evidence, policymakers and service providers may underestimate the scope of the problem or fail to tailor interventions appropriately.

This knowledge gap is further reinforced by dominant public health approaches that frame FI narrowly as a nutritional or food access issue, often overlooking its broader psychosocial dimensions. Many studies on migrant health have focused on undernutrition (e.g., micronutrient deficiencies or underweight) in resettled communities [24,25]. Conversely, other research has focused on emerging issues of overnutrition and diet-related chronic diseases [26–28]. While these perspectives are essential, they often fail to address the lived experience of food insecurity and its psychological ramifications. Anthropological research has long underscored that FI is not just about caloric insufficiency or hunger; it also entails loss of dignity, erosion of autonomy, and challenges to culturally appropriate food practices [29,30]. In migrant and refugee contexts, these less tangible aspects of FI, such as being unable to provide traditional foods for one's family or the humiliation of depending on food aid, can be deeply intertwined with mental well-being. Ignoring these psychosocial elements in public health discourse means overlooking a key reason why FI can lead to emotional distress.

Given these gaps in knowledge and perspective, the mental health impacts of FI among migrants and refugees in HICs remain poorly understood. This lack of understanding limits the development of effective interventions and policies. It is therefore essential to determine which risk factors most significantly affect the mental health of displaced populations to prioritize and tailor support programs. To address this need, the present study undertakes a systematic review and meta-analysis to examine the association between food insecurity and mental health outcomes, specifically depression, anxiety, and stress, among migrants and refugees in high-income countries. By synthesizing evidence across diverse studies, we aim to fill this critical knowledge gap and provide clearer insights into how FI influences psychological

well-being in these communities. The findings aim to inform culturally responsive interventions and policies that address the interconnected nutritional and mental health needs of migrants and refugees. Addressing FI-related mental health disparities among displaced populations is not only a clinical and public health imperative; it also aligns with global commitments, including the United Nations Sustainable Development Goals (SDGs 1, 2, 3, and 10). These goals aim to reduce poverty and hunger, improve health and well-being, and reduce inequalities.

## 2. Methods

This review is reported in accordance with the Preferred Reporting Items for Systematic Reviews and Meta-Analyses (PRISMA) guidelines [31]. The PRISMA 2020 checklist for the manuscript is provided in S1 Checklist, and the PRISMA 2020 for abstract checklist is provided in S2 Checklist. Moreover, the protocol for this systematic review has been published [32] (see S1 File) and registered with the PROSPERO International Prospective Register of Systematic Reviews website (http://www.crd.york.ac.uk/PROSPERO) (PROSPERO registration number = CRD42024525690) [33].

### Eligibility criteria

The screening process was guided by predefined eligibility criteria and structured according to the PICO framework (Population, Intervention/Exposure of interest, Context, and Outcome) to define the boundaries of the research question clearly. We included quantitative observational studies (ecological, cross-sectional, case-control, and cohort) conducted in high-income countries that reported an effect estimate (or sufficient data to derive one) for the association between food insecurity and depression, anxiety, and/or stress among migrants or refugees. Although the search strategy encompassed all observational study designs, the final set of eligible studies was largely cross-sectional (Table 1).

### Information sources

We conducted a comprehensive search of multiple electronic databases to identify relevant observational studies on food insecurity and mental health in migrant and refugee populations. Specifically, we searched MEDLINE (via Ovid), Embase (via Ovid), Scopus, Web of Science, and PsycINFO (via Ovid) for articles published from January 1, 2008, up to our final search date in late 2025. These databases were selected for their comprehensive coverage of medical, public health, psychological, and social science literature, ensuring that we captured the vast majority of studies relevant to our topic. The start date of 2008 was selected to encompass approximately 15 + years of research, a period during which attention to food insecurity in high-income settings increased. For each database, we tailored the platform-specific search interface (e.g., Ovid Medline) to execute our queries. In developing the search strategy, we consulted with an experienced research librarian to refine keywords and indexing terms. Our search terms combined concepts of "food insecurity" with "migrant" or "refugee" and mental health-related terms (e.g., "psychological distress", "mental well-being"), along with

**Table 1. PICOS Criteria for Inclusion of Studies.**

| Mnemonic | Adapted Parameter | Criterion |
|---|---|---|
| P | Population or Participants | Migrants and refugees aged ≥18 years or older living in high-income countries. Excludes general populations and children/adolescents (<18 years). |
| I/E | Phenomena of Interest | Food insecurity is measured using validated scales (e.g., HFIAS, FIES). |
| C | Context | High-income countries (classified as high- or very-high human development index). |
| O | Outcomes | Mental health outcomes, including depression, anxiety, and stress, were measured by validated instruments (e.g., K6, K10, DASS-21, GHQ). |

relevant synonyms. We applied these terms uniformly across all sources to maximize consistency. In addition to database searches, we manually examined the reference lists of all included observational studies and relevant review articles to identify additional studies that our database searches might have missed. We also performed citation tracking (both forward and backward) using Google Scholar and Web of Science, checking which newer articles had cited our included studies. These supplementary search techniques (sometimes called "snowballing") served as a cross-check to ensure saturation of the literature. However, these efforts did not yield any additional eligible studies beyond those identified through the database search. All searches and information sources, including dates of the last search, are documented to uphold reproducibility. We did not conduct a systematic grey literature search (e.g., theses, reports), which may have limited the capture of unpublished evidence.

We limited the search to English-language publications, a decision made after careful consideration of the potential for language bias. While including studies in all languages can, in theory, capture a broader evidence base, empirical research indicates that excluding non-English studies is unlikely to distort review findings. Multiple meta-analyses have found that language-restricted reviews typically reach the same conclusions as unrestricted reviews. For example, previous analyses found no significant difference in overall effect estimates between English-only meta-analyses and those that included other languages [34,35]. More recently, Nussbaumer-Streit et al. (2020) demonstrated that omitting non-English publications rarely changes the direction or significance of results [36]. In their sample of Cochrane reviews, the inclusion of other languages would not have altered a single conclusion. We acknowledge the general risk of language bias, recognizing that excluding non-English studies may skew the evidence if, for instance, essential data were available only in another language. However, given the context of our review, we believe this risk is minimal. Research on migrant food insecurity in high-income countries is predominantly disseminated in English-language journals, and any non-English studies in this niche are unlikely to be numerous enough to alter the observed trends. Moreover, the evidence cited above suggests that even when non-English studies are available, excluding them does not significantly affect meta-analytic outcomes. On a practical note, resource constraints (in terms of translation capacity) also informed our English-only approach. We made this choice transparently, balancing inclusivity with feasibility, and based on verified methodological evidence that it would not compromise the integrity of our review's conclusions. In summary, by restricting our analysis to English-language sources, we maintained focus on the core literature without sacrificing the validity of our results. We have noted this limitation, providing supporting justification for compliance with best-practice guidelines and reassuring readers that our findings are robust against concerns of language bias.

## Search strategy

We developed the search strategy around four key concepts: population, exposure, outcomes, and setting, using a combination of controlled vocabulary and free-text terms for each concept. Controlled vocabulary terms were drawn from the standardized indexing in each database (e.g., Medical Subject Headings [MeSH] in MEDLINE/PsycINFO and Emtree in Embase) to ensure the precise retrieval of concept-specific literature. For instance, we incorporated MeSH terms such as "Refugees" and "Emigration and Immigration" to represent the population concept, "Hunger" and "Food Supply" for the exposure concept, "Depression", "Anxiety","Stress" and "Mental Health" for the outcomes concept, and "Developed Countries" for the setting. A broad range of free-text keywords complemented these controlled terms to capture variations and synonyms. For example, the population concept was also searched with terms like "migrant*", "refugee*", and "asylum seek*"; the exposure concept with terms such as "food insecur*" and "hunger"; the outcomes concept with terms including "depress*", "anxiet*", "psychological distress", and "mental health"; and the setting concept with terms for high-income settings like "high-income countr*" and "developed countr*", as well as names of specific high-income countries or regions as needed.

Boolean operators and other search refinements were used to structure the strategy. Within each concept category, we combined terms using the OR operator (to broaden the search to include any of the synonyms within that concept). Then

we used AND across the four concept categories (to ensure that retrieved records included at least one term from each idea, thereby focusing the results on our review's intersection of interest). We applied truncation (wildcard symbols) where appropriate to capture multiple word endings, for example, using "migrant*" retrieves migrant, migrants, migration, etc. We also applied filters to limit the results by language (English only), publication type (restricting to peer-reviewed journal articles), and publication date (2008–2025) in line with our inclusion criteria. These limits were implemented either within the database search filters or during the screening process, as appropriate to each database's capabilities.

The search strategy was carefully tailored to each database to accommodate differences in search syntax and indexing systems. Database-specific field codes and subject headings were used when available: for example, in MEDLINE and PsycINFO (via Ovid), we used field tags for MeSH terms and keywords, in Embase we used Emtree terms alongside keywords, and in Scopus and Web of Science (which lack a controlled vocabulary system) we relied on title/abstract keyword searches only. Minor adjustments (e.g., syntax for truncation or phrase searching) were made for each platform to ensure that the strategy was executed correctly across all databases. The complete search strategies for all databases are provided in the supplementary materials (see S1 Appendix for the full search strings). An example is the detailed MEDLINE (Ovid) search string included in the supplement, which illustrates how MeSH terms for migrants/refugees, food insecurity/hunger, mental health outcomes, and high-income countries were combined with corresponding keywords using the OR/AND structure described. Similar search logic was applied in the other databases, with adjustments as needed to conform to each database's search interface. The full database search strategies are provided in S1 Appendix.

## Study selection

The study selection process began with removing duplicate records from the combined search results using Covidence software. Three independent reviewers (RB, KA, and KE) then screened the titles and abstracts of all remaining articles, applying the predefined inclusion criteria (population, exposure, context, outcome, setting, and study design). Articles deemed potentially relevant were retrieved in full text for a more thorough evaluation. Any discrepancies during this initial screening were resolved through group discussion or, when necessary, by consulting a third reviewer (KE and AA). Subsequently, the same reviewers independently assessed each full-text article against the eligibility criteria. Reasons for exclusion were documented to ensure transparency and replicability.

## Data abstraction and data items

Data extraction was conducted using a standardized abstraction form that captured key characteristics of each study, including time frame, study design, sampling method, data collection procedures, and participant demographics. Two independent reviewers completed the data abstraction spreadsheets, which were cross-verified to minimize errors and resolve discrepancies. The form also included detailed descriptions of exposure and outcomes, specifying the measurement instruments (e.g., validated scales or rapid screening tools) and the methodological approaches used to assess food insecurity and mental health outcomes. Where available, preliminary information on the number of participants in each exposure and outcome group was also recorded.

Exposure and outcome definitions were condensed and consolidated into a results table to enhance comparability and facilitate quality assessment, allowing for clear cross-study comparisons. A pilot test of the data abstraction form was conducted on a small purposive sample of articles to ensure that all relevant variables were accurately captured. During this pilot phase, the abstraction tool was iteratively refined to address any ambiguities and improve its comprehensiveness. Finally, studies were grouped according to population characteristics and outcome measures (e.g., depression, anxiety, stress), enabling a structured quality assessment and a synthesized presentation of findings in subsequent analyses.

## Quality assessment

All included analytical cross-sectional studies were assessed using the Joanna Briggs Institute (JBI) Critical Appraisal Checklist for Analytical Cross-Sectional Studies, which comprises eight core items addressing study population selection, exposure measurement, control of confounding factors, and outcome assessment. This approach offers a robust framework for evaluating whether each study's methods are sufficiently rigorous to support valid inferences about the association between food insecurity (exposure) and psychological distress (outcome).

Each item was scored as "Yes" (1 point) or "No/Unclear" (0 points), producing a total possible score ranging from 0 to 8. Following established guidelines, studies with scores of 7–8 were considered high quality, those with scores of 4–6 were deemed moderate quality, and those with scores of 0–3 were classified as low quality. Two independent reviewers applied the checklist to each study; any scoring discrepancies were resolved through discussion or, if necessary, with the input of a third reviewer (KE and AA). Only studies reporting sufficient detail on sample recruitment, measurement validity, confounding, and statistical analyses were retained to ensure methodological rigor. This quality appraisal process helps uphold the credibility of our findings by ensuring that the evidence synthesized is based on methodologically sound research.

## Data synthesis

We conducted a meta-analysis of observational studies to examine the relationship between food insecurity (FI) and mental health outcomes (depression, anxiety, and stress) among migrants and refugees living in high-income countries. From each eligible study, we extracted the adjusted odds ratios (ORs) and 95% confidence intervals (CIs) for the associations of interest, prioritizing the most fully adjusted estimates (i.e., those controlling for the most significant number of relevant confounders). If a study reported an effect size in a form other than ORs (such as a regression coefficient or mean difference), we converted these estimates to ORs with corresponding 95% CIs to maintain a consistent effect measure across all studies. We also standardized effect directions by aligning reference categories. If different studies used different baseline groups (e.g., some using male as the reference for gender and others using female), we inverted the ORs and CIs as necessary to ensure a common reference category was used across studies. This approach ensured that groups such as females (gender), low-income individuals (household income), and individuals with poor health status (general health) consistently served as the comparison group in our analysis. All such data transformations and realignments were thoroughly documented to ensure transparency and replicability.

We pooled the extracted effect estimates using a DerSimonian–Laird random-effects model, which accounts for between-study variability in the actual effect sizes. Meta-analyses were conducted whenever two or more studies provided data suitable for pooling on a given outcome or subgroup. For each outcome, we combined the ORs (and CIs) from individual studies using this random-effects approach, thereby incorporating both within-study and between-study uncertainty into the combined estimate. We also generated forest plots to visualize the individual study results and the pooled estimates with their CIs.

Subgroup analyses were planned a priori to explore potential moderators of the FI–mental health association, based on factors identified in the literature as possible influences on FI experiences or mental health. Specifically, we stratified analyses by gender (female vs. male), marital status (single vs. married), general self-rated health (poor vs. good), and household income-to-poverty ratio (lower vs. higher). In cases where a study reported more than two categories for one of these variables (for example, multiple income brackets or a detailed health status scale), we collapsed the categories into binary groups for consistency (e.g., combining moderate and high-income categories into "high income" vs. "low income," or grouping various levels of less-than-good health into a single "poor health" category vs. good health). When subgroup-specific estimates were available (either reported directly by the study or derived from stratified data), we performed separate meta-analyses within each subgroup. This allowed us to determine, for instance, whether the association between FI and mental health differed for women compared to men, or lower-income households compared to higher-income ones.

We assessed heterogeneity among studies using Cochran's Q test and the I² statistic. We interpreted I² values of approximately 25%, 50%, and 75% as indicative of low, moderate, and high heterogeneity, respectively. A Cochran's Q test with p < 0.10 and I² value greater than 50% was considered to reflect substantial between-study heterogeneity. In the presence of notable heterogeneity, we further investigated potential sources by conducting additional subgroup analyses (as described above) and sensitivity analyses. Sensitivity analyses included examining the influence of individual studies on the pooled result (by omitting one study at a time) and restricting analyses to subsets of studies (e.g., only those with low risk of bias or only those focusing on a specific region) to assess whether the findings remained robust. Finally, we evaluated publication bias using visual and statistical methods. We created funnel plots for each meta-analysis to visually assess asymmetry that might indicate selective publication of results. Additionally, we performed Begg's and Egger's tests to determine funnel plot asymmetry, with p-values < 0.05 indicating potential publication bias. If evidence of such bias was present, we planned to apply the trim-and-fill method to estimate the number of missing studies and adjust the pooled effect estimate accordingly, yielding a bias-corrected result. All analyses were performed using Stata 18.0, and all plots (forest plots for meta-analyses and funnel plots for publication bias assessment) were generated in Stata.

## 3. Results

The literature search yielded 758 unique records from the five databases: Medline (79), Scopus (128), Web of Science (193), PsycINFO (144), and Embase (214). After removing 332 duplicates, 426 titles and abstracts were screened; 309 records were excluded for failing to meet the inclusion criteria (most commonly due to an incorrect population or the absence of an FI measure). We assessed 117 full-text articles in detail, excluding 100 for various reasons (e.g., ineligible population, outcomes, or insufficient data). Ultimately, 17 studies met all criteria and were included in the systematic review and meta-analyses (Fig 1).

### Characteristics of the study

All 17 studies meeting the inclusion criteria employed an observational study design, with one notable exception that incorporated an exploratory mixed-methods component. Geographically, the majority of studies were conducted in North America (64.7%), including eight from the United States and three from Canada. Europe contributed five studies (29.4%), comprising three from Norway, one from France, and one from Sweden. Additionally, one study (5.9%) originated from Israel, representing the Middle East. Key study characteristics are summarised in Table 2. Across included studies, reporting of funding sources was inconsistent, which limited our ability to describe funding patterns or assess funding-related differences.

Sample sizes varied substantially, ranging from 90 to 3,273,493 participants, with a total of approximately 4.3 million individuals across the studies. Aggregated data revealed a nearly balanced gender distribution, comprising 2,111,445 males (49.2%) and 2,178,883 females (50.8%). Participants' reported ages ranged from 30.6 to 49.74 years, indicating a predominantly adult population in early to middle adulthood.

Regarding the measurement of food insecurity (FI), 12 studies (70.6%) used validated scale-based instruments, primarily adaptations of the USDA Household Food Security Survey Module, whereas five studies (29.4%) employed rapid assessment tools. Psychological distress outcomes were evaluated using several validated instruments, including the Hopkins Symptom Checklist (HSCL-10 or HSCL-25), Kessler scales (K6/K10), Patient Health Questionnaire (PHQ-9, PHQ-2, GAD-2), Depression Anxiety Stress Scales (DASS-21), Center for Epidemiologic Studies Depression Scale (CES-D), Beck Depression Inventory-II (BDI-II), PTSD Checklist, World Health Organization Composite International Diagnostic Interview (WHO-CIDI), and Harvard Program in Refugee Trauma Depression Scale (HPRTDS).

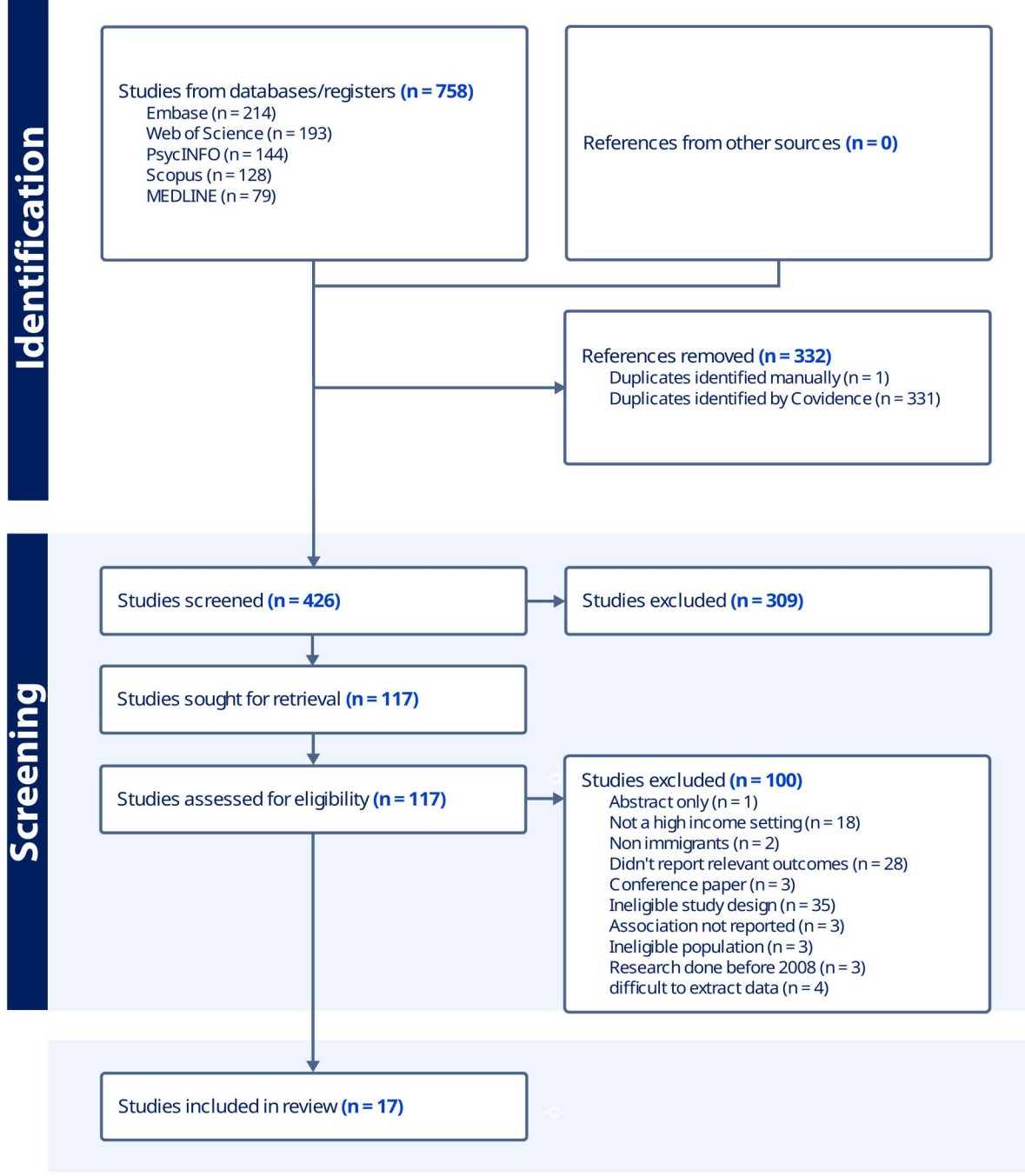

**Fig 1. PRISMA flow diagram of study selection.** PRISMA, Preferred Reporting Items for Systematic Reviews and Meta-Analyses.

## Quality assessment

All included studies underwent rigorous methodological appraisal using the Joanna Briggs Institute (JBI) Critical Appraisal Checklist for Analytical Cross-Sectional Studies. The JBI checklist comprises eight critical methodological domains: clearly

**Table 2. Characteristics of the Studies.**

| Authors/Years | Study Design | Host Country | Popula-tion size | Mean age | Gender | | FI Measurement | Psychological Distress Assessment | Quality Assessment |
|---|---|---|---|---|---|---|---|---|---|
| | | | | | Male N(%) | Female N(%) | | | JBI |
| Zangiabadi et al. (2024) | Cross-sectional | Canada | 540 | 39.77 years | 211 (39.07%) | 329 (60.93%) | Rapid measure | DASS-21 | High |
| Sonia Lai (2020). | Cross-sectional | USA | 2095 | 47.78 years | 1,008 (48.07%) | 1,087 (51.93%) | Rapid measure | WHO-CIDI | High |
| Hadley et al. (2008). | Cross-sectional | USA | 431 | 32 | 298 (69.6%) | 130 (30.4%) | Scale measure | Self-reported mental health status in the last 30 days | High |
| Talham (2023) | Cross-sectional | USA | 5310 | | 2331 (43.9%) | 2979 (56.1%) | Scale measure | PHQ-2 and GAD-2 scales | High |
| Adzrago (2023) | Cross-sectional | USA | 4320 | Not Specified | 1922(47.83%) | 2398(52.17%) | Scale measure | Kessler 6 scale | High |
| McKinney et al (2019). | Cross sectional | USA | 61274 | Not Specified | 30005 (48.96%) | 31269 (51.03)% | Scale measure | Kessler 6 scale | High |
| Peterman (2012) | Cross-sectional | USA | 150 | 46.5 years | 0 | 150(100%) | Scale measure | HPRTDS | High |
| Islam (2014) | Cross-sectional study | Canada | 997,706 | 41.85 | 530,978 (53.2%) | 466,728 (46.8%) | Scale measure | Self-reported clinically diag-nosed and self-perceived variables | High |
| Pulgar (2015) | Cross-sectional | USA | 248 | 30.6 years | | 248(100%) | Scale measure | CES-D | High |
| Myhrvold and Milada (2017) | Exploratory mixed methods | Norway | 90 | | 51 (57%) | 39 (43%) | Scale measure | HSCL-25 | High |
| Islam (2018) | Cross-sectional | Canada | 3,273,493 | 49.74 years | 1,574,733 (48.11%) | 1,698,760 (51.89%) | Rapid measure | Past-year mental health consultation | High |
| Andersson (2018) | Cross-sectional | Sweeden | 104 | 31.28 years | 48 (54.55%) | 40 (45.45%) | Scale measure | BDI-II, BAI, PTSD Checklist (PCL-5) | High |
| Myhrvold and Milada (2019) | Cross-sectional | Norway | 90 | NA | 51 (57%) | 39 (43%) | Scale measure | HSCL-25 | High |
| Attal (2020) | Cross-sectional | Israel | 307 | 37 years | 67 (21.8%) | 240 (78.2%) | Rapid measure | HSCL-10 | High |
| Yueqi Li (2022). | Cross-sectional | USA | 6,857 | 44.5 years | 3,610 (52.6%) | 3,247 (47.4%) | Scale measure | PHQ-9 | High |
| Richard et al. (2021). | Cross-sectional | France | 551 | 38.42 years | 271 (49.18%) | 280 (50.82%) | Scale measure | Kessler 6 scale | High |
| Kamelkova (2023) | Cross-sectional | Norway | 353 | 35.6 years | 169 (48.3%) | 181 (51.7%) | Scale measure | HSCL-10 | High |

*Abbreviations: BAI, Beck Anxiety Inventory; BDI-II, Beck Depression Inventory-II; CES-D, Center for Epidemiologic Studies Depression Scale; DASS-21, Depression Anxiety Stress Scales-21; FI, Food Insecurity; GAD-2, Generalized Anxiety Disorder-2; HPRTDS, Harvard Program in Refugee Trauma Depression Scale; HSCL-10/HSCL-25, Hopkins Symptom Checklist (10-item/25-item); JBI, Joanna Briggs Institute; NA, Not Available; PCL-5, PTSD Checklist for DSM-5; PHQ-2/PHQ-9, Patient Health Questionnaire (2-item/9-item); PTSD, Post-Traumatic Stress Disorder; WHO-CIDI, World Health Organization Composite International Diagnostic Interview.*

defined inclusion criteria, detailed description of study subjects and settings, valid and reliable exposure measurement (e.g., food insecurity), standardized outcome assessment (e.g., psychological distress), identification and adjustment for potential confounding factors, valid and reliable outcome measurement, and appropriate statistical analysis.

All 17 studies received high-quality ratings, meeting 7 or 8 of the 8 possible JBI criteria (See Table 2 below and the S1 Table, for study-level appraisal details). The two domains most likely to introduce bias were those related to confounding, because cross-sectional designs limit causal inference, and some studies adjusted for a restricted set of covariates. In addition, a minority

of studies relied on self-reported measures of food insecurity and mental health outcomes, which may introduce measurement or reporting bias; however, these measures were generally validated instruments, reducing this risk. Quality assessments were independently conducted by two reviewers, with discrepancies resolved through consensus or adjudication by a third reviewer. The consistently high methodological quality across all studies significantly enhances the reliability and validity of the meta-analytic findings, supporting strong and credible associations between food insecurity and psychological distress outcomes.

## Meta-analyses

**Depression.** As illustrated in Fig 2, the pooled analysis using a random-effects model indicated that food insecurity (FI) significantly increased the odds of depression among migrants and refugees in high-income countries (adjusted odds ratio (aOR) = 1.44; 95% CI: 1.32–1.57; $p < 0.001$). Substantial heterogeneity was detected among the included studies ($I^2 = 91.4\%$, $p < 0.001$). Additionally, the funnel plot indicated potential publication bias or small-study effects, which were statistically confirmed by Begg's test (Kendall's tau = 0.368, $p = 0.042$) (see S1 Fig).

To explore sources of heterogeneity, subgroup analyses were performed by gender (female vs. male participants), marital status (unmarried vs. married), general health status (poor vs. good self-rated health), and household income-to-poverty ratio (lower vs. higher income relative to needs). The analyses demonstrated significantly higher odds of depression among food-insecure females compared to males (aOR = 1.44; 95% CI: 1.32–1.57; $I^2 = 78.6\%$, $p < 0.001$) (Fig 3A). Similarly, unmarried individuals experiencing FI exhibited consistently higher depression risk compared to married individuals (aOR = 1.70; 95% CI: 1.49–1.83; $I^2 = 0.0\%$, $p = 0.569$) (Fig 3B). Furthermore, migrants and refugees reporting poor general health had substantially increased odds of depression (aOR = 3.60; 95% CI: 2.97–4.36; $I^2 = 7.6\%$, $p = 0.298$) compared to those with good general health (Fig 3C).

Subgroup analysis based on the family income-to-poverty ratio indicated that migrants and refugees with a higher household income relative to the poverty line (> one poverty line threshold) had significantly lower odds of depression compared to those below this threshold (OR = 0.90; 95% CI: 0.84–0.98; $I^2 = 0.0\%$, $p = 0.007$) (Fig 3D).

**Anxiety.** As shown in Fig 4, the pooled analysis revealed that food-insecure migrants and refugees had more than twice the odds of experiencing anxiety compared to their food-secure counterparts (aOR = 2.3; 95% CI: 1.85–2.88; $p < 0.001$). Begg's test indicated no significant evidence of publication bias ($p = 0.719$) (see S2 Fig). However, considerable heterogeneity was observed across studies ($I^2 = 67.0\%$, $p = 0.01$), prompting further subgroup analyses stratified by general health status and gender to explore potential sources of variability.

Fig 5 illustrates subgroup analysis by general health and gender. The subgroup analysis by general health showed that migrants and refugees facing both food insecurity and poor self-rated health had nearly four times greater odds of anxiety (aOR = 3.7; 95% CI: 2.93–4.66; $I^2 = 0.0\%$; $p = 0.954$) compared to those with good self-rated health (Fig 5B), with no observed heterogeneity. In contrast, subgroup analyses by gender (Female) (aOR = 0.2; 95% CI: 0.21–0.27; $I^2 = 99.4\%$; $p < 0.001$) (Fig 5A) revealed substantial heterogeneity and no consistent patterns, suggesting that gender did not significantly modify the association between food insecurity and anxiety in this population.

**Stress.** Fig 6 illustrates the robust association between food insecurity (FI) and stress among migrants and refugees in high-income countries, as evidenced by observational studies conducted between 2014 and 2025. The pooled inverse-variance-weighted odds ratio (aOR) of 5.9 (95% CI: 2.95–11.94, $p < 0.001$) indicates that migrants and refugees experiencing food insecurity have significantly higher odds of stress than their food-secure counterparts. The analysis showed complete homogeneity across studies ($I^2 = 0.0\%$, $p = 0.89$), supporting the reliability and consistency of this relationship.

Fig 7 illustrates subgroup analysis by gender. The subgroup analysis by gender showed no significant association between gender and stress among food-insecure migrants and refugees (aOR = 1.1; 95% CI: 0.32–3.72; $I^2 = 76.6\%$, $p = 0.039$) (Fig 7). We did not conduct further subgroup analyses to determine the association between FI and stress risk due to the limited number of studies.

**Sensitivity analysis and publication bias.** A sensitivity analysis was conducted to assess the impact of the Andersson (2018) study on the overall results, given its notably wide confidence interval, which may indicate potential instability or lower precision (Table 3).

**Fig 2. Association Between Food Insecurity and Depression.** Forest plot of odds ratios (ORs) for depression comparing food-insecure versus food-secure migrants and refugees in high-income countries. Squares indicate study-specific ORs with 95% confidence intervals (CIs); the diamond indicates the pooled OR (random-effects model). I² denotes between-study heterogeneity.

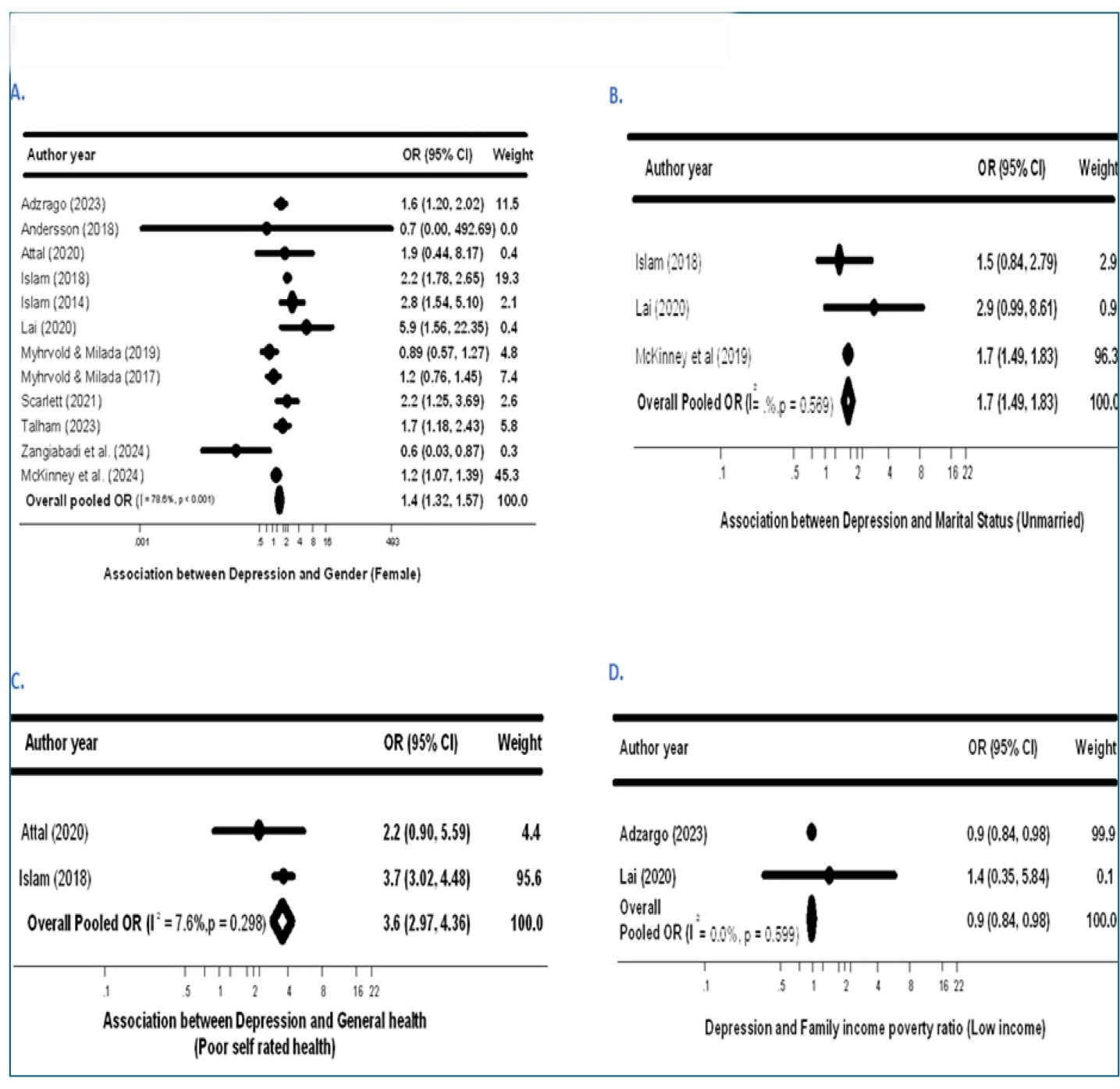

**Fig 3. Additional factors influencing the association between Food Insecurity and depression.** Subgroup forest plots for (A) gender (female), (B) marital status (unmarried), (C) general health (poor self-rated health), and (D) family income-to-poverty ratio (low income). Squares indicate study-specific ORs with 95% CIs; diamonds indicate pooled effects (random-effects model). I² denotes heterogeneity.

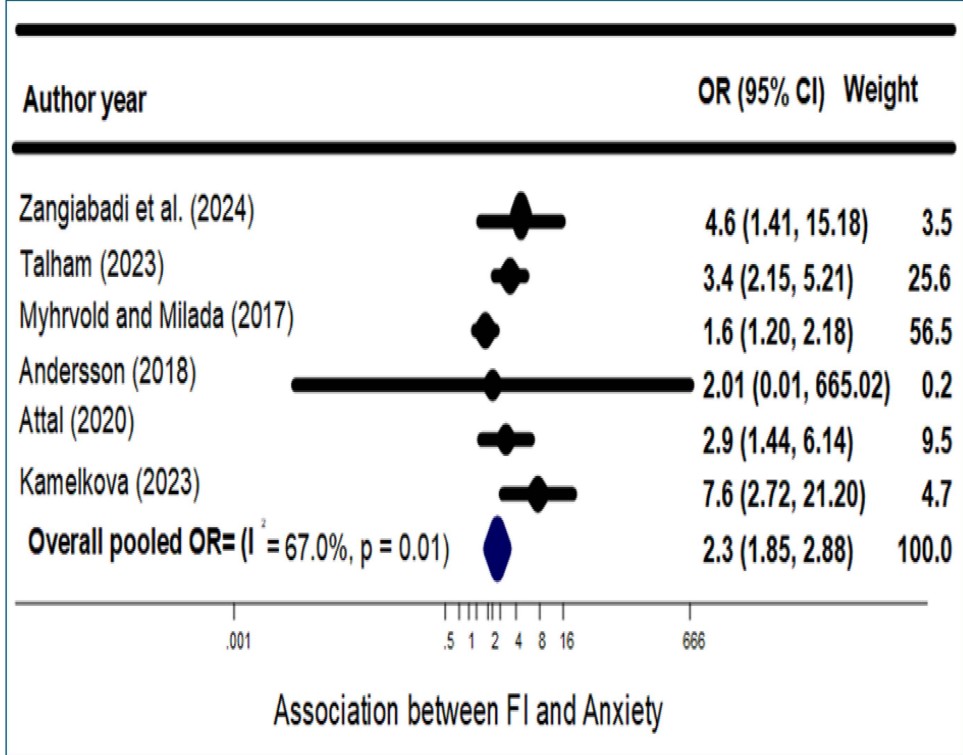

**Fig 4. Association between FI and Anxiety.** Forest plot of odds ratios (ORs) for anxiety comparing food-insecure versus food-secure migrants and refugees. Squares indicate study-specific ORs with 95% confidence intervals (CIs); the diamond indicates the pooled OR (random-effects model). FI, food insecurity; I² denotes heterogeneity.

After removing the Andersson (2018) study, the results remained consistent, confirming the robustness of the pooled odds ratio (OR). The overall OR stayed stable at around 1.40–1.41, and importantly, the recalculated confidence intervals continued to exclude 1, indicating persistent statistical significance. This stability suggests that the results of this particular study did not influence the direction or significance of the pooled results.

The magnitude of the pooled OR changed very slightly (approximately 0.01), underscoring the strong consistency of the findings. This stability adds confidence to the conclusion that the relationship between food insecurity and depression is not disproportionately influenced by individual studies such as Andersson (2018) but instead represents a consistent pattern observed across multiple studies.

In the gender-specific analyses examining depression, anxiety, and food insecurity, the sensitivity analysis also showed stable results after removing the Andersson study. The pooled OR values for gender were either unchanged or changed negligibly, and the confidence intervals consistently excluded 1. This further highlights the reliability of these findings.

In summary, the leave-one-out sensitivity analysis indicates that excluding the Andersson (2018) study does not significantly alter the meta-analytic results. The pooled odds ratio (OR) remains consistently significant and in the same direction across all analyses. Although the observed heterogeneity warrants further exploration, perhaps through subgroup analyses or meta-regression, it is not driven by any single study. This enhances the overall credibility and reliability of the meta-analytic conclusions.

## 4. Discussion

This review synthesises evidence on the association between food insecurity and three commonly reported mental health outcomes, depression, anxiety, and stress, among migrants and refugees living in high-income

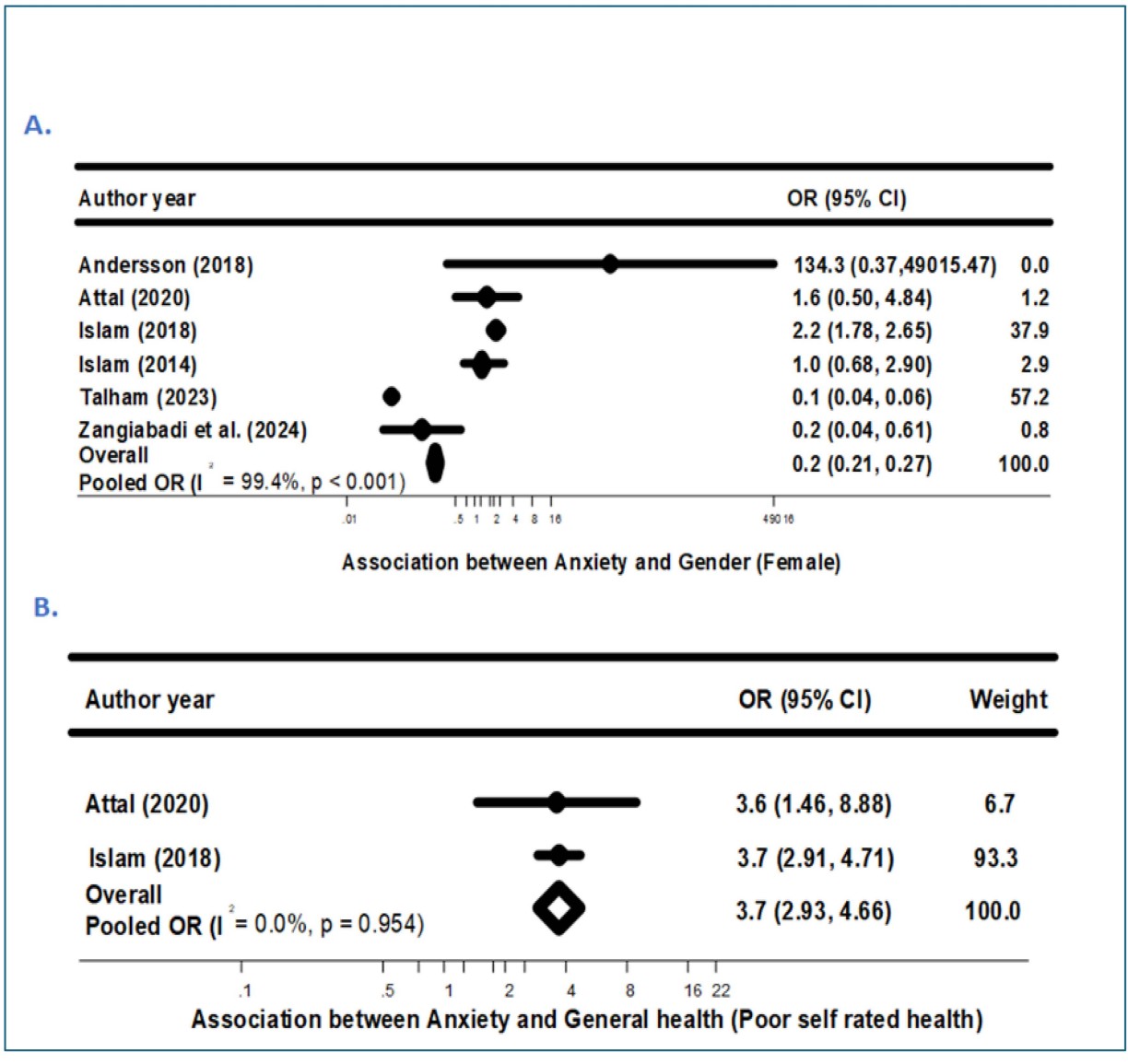

**Fig 5. Additional factors influencing the association between Food Insecurity and anxiety.** Subgroup forest plots for (A) gender (female) and (B) general health (poor self-rated health). Squares indicate study-specific ORs with 95% confidence intervals (CIs); diamonds indicate pooled effects (random-effects model). FI, food insecurity; I² denotes heterogeneity.

countries. We first summarise the pooled associations, then discuss potential mechanisms and subgroup patterns (e.g., age, marital status, gender, health status, and socioeconomic indicators), before outlining limitations of the underlying observational evidence and implications for policy, screening, and targeted support. In summary, this systematic review and meta-analysis show that food insecurity among migrants and refugees in high-income countries is linked to higher odds of depression, anxiety, and stress. The association was stronger among women, unmarried individuals, and those with poor self-rated health, while higher household income appeared protective. Overall, the findings underscore the multifactorial ways food insecurity affects mental health in these communities.

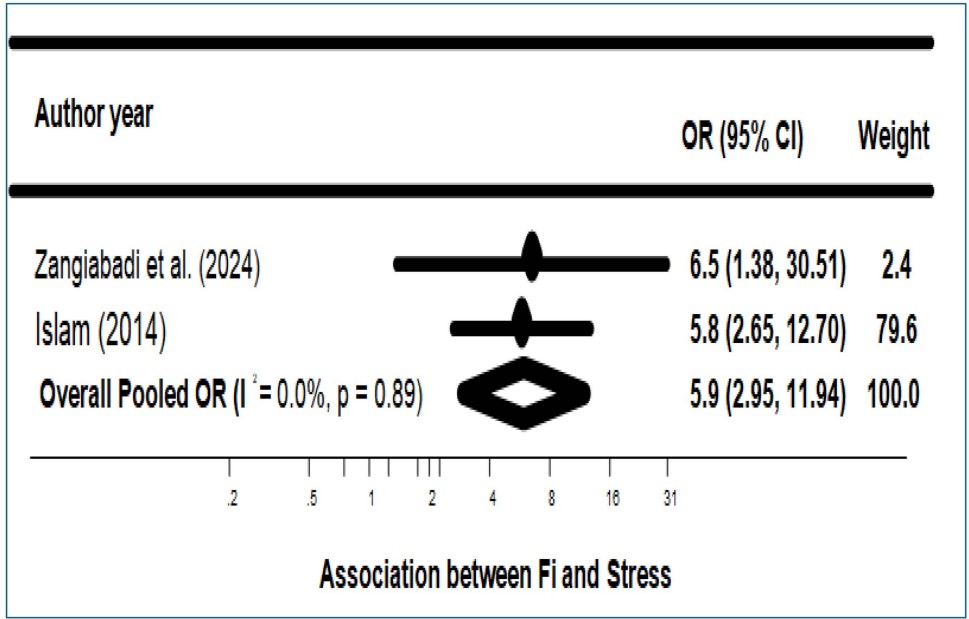

**Fig 6. Association between Food Insecurity and Stress.** Forest plot of odds ratios (ORs) for stress comparing food-insecure versus food-secure migrants and refugees. Squares indicate study-specific ORs with 95% confidence intervals (CIs); the diamond indicates the pooled OR (random-effects model). FI, food insecurity; I² denotes heterogeneity.

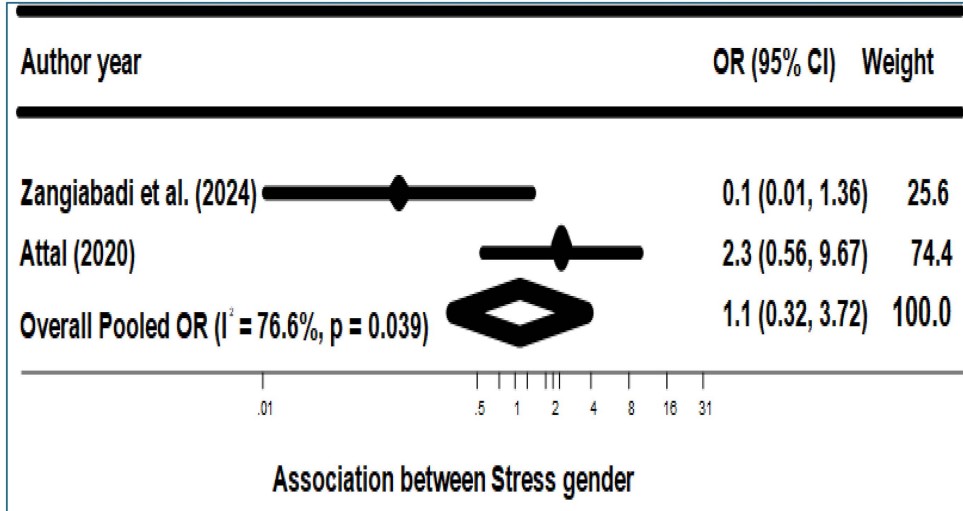

**Fig 7. Association between Stress and gender.** Subgroup forest plot for stress by gender (female). Squares indicate study-specific ORs with 95% confidence intervals (CIs); the diamond indicates the pooled OR (random-effects model). I² denotes heterogeneity.

Specifically, our findings on depression reinforce prior evidence from various population-based studies, including research among low-income and marginalized groups in developed nations, which consistently highlight FI as a critical determinant of depressive symptoms [37,38]. The chronic uncertainty associated with food access, feelings of deprivation, and societal stigma surrounding FI can induce sustained emotional distress, diminished self-worth, and social isolation,

**Table 3. Sensitivity analysis summary table of depression, anxiety, gender, and FI.**

| Meta-analyses graphs | With Andersson (2018) | Without Andersson (2018) |
|---|---|---|
| | AOR (95%CI) | AOR (95%CI) |
| Association between FI and depression | 1.40 (1.32-1.57) | 1.41 (1.33-1.59) |
| Association between depression and Gender (female) | 1.40 (1.32-1.57) | 1.41 (1.3-1.6) |
| Association between FI and Anxiety | 2.3 (1.85-2.88) | 2.3 (1.84-2.88) |
| Association between Anxiety and Gender (female) | 0.24 (0.21-0.27) | 0.24 (0.21-0.27) |

all central elements in the pathogenesis of depression [19,39]. Additionally, biologically, persistent nutritional deficiencies associated with FI negatively impact neurological processes, impair cognitive functioning, and reduce mood stability, exacerbating depression risks [40,41].

Concerning anxiety, our findings confirm a strong association between food insecurity (FI) and heightened anxiety symptoms. This outcome aligns strongly with prior studies indicating anxiety symptoms arise predominantly from prolonged uncertainty and persistent concerns about food adequacy [15,42]. This link likely reflects migrants' and refugees' ongoing concern with securing food, combined with broader anxieties about economic survival, social integration, and family well-being in new and unfamiliar environments [21]. Anxiety related to FI, especially among displaced populations, may be particularly pronounced due to limited social support, institutional barriers, and systemic marginalization [19].

Similarly, FI was strongly associated with elevated stress levels in our study, corroborating earlier qualitative and quantitative research highlighting chronic stress as a profound consequence of FI [39,43]. The persistent insecurity and unpredictability associated with obtaining adequate food can cause chronic stress, further exacerbating physical and mental health disparities among vulnerable populations. Indeed, chronic stress induced by FI can activate physiological stress pathways, negatively influencing long-term physical health and perpetuating cycles of psychological distress [15].

Beyond these direct effects, several demographic factors emerged as significant moderators of the FI–mental health relationship, particularly for depression. Gender proved to be a powerful influence: food-insecure women experienced markedly higher levels of depression than their male counterparts, echoing prior evidence that women bear a disproportionate psychological burden from FI [22,44]. This heightened vulnerability among women is often attributed to structural gender inequalities, economic precarity, and caregiving responsibilities that fall disproportionately on women. Marital status likewise played a role in depression outcomes: unmarried individuals (i.e., those without partners) exhibited more severe FI-linked depression than did married individuals, consistent with the protective effects of spousal support. Having a partner can buffer against the psychological hardships of FI, whereas social isolation and lack of emotional support heighten feelings of helplessness in unmarried migrants and refugees [21,43]. Notably, neither gender nor marital status showed any significant moderating effect on anxiety or stress in our subgroup analyses. This absence of effect in the other mental health domains could reflect limited power (due to fewer studies) or high outcome variability, underscoring the need for further research into how gender and social support influence anxiety and stress under FI conditions.

Physical health status was another critical moderator in the FI–mental health nexus. Our results indicate that individuals with poorer general health experienced disproportionately higher levels of both depression and anxiety when facing FI. This pattern reinforces the well-documented bidirectional link between physical and mental health: chronic physical ailments can amplify psychological distress, and conversely, mental strain can worsen physical health outcomes. In the context of FI, those with compromised health likely face compounded anxiety and depressive symptoms as they struggle

with the dual challenges of securing food and managing ongoing health concerns [15,42]. These findings underscore the importance of integrated interventions that simultaneously address physical health and provide psychological support to mitigate the mental health burden of FI.

Among the socioeconomic factors examined, our analysis highlighted the moderating role of household income. Specifically, income significantly moderated the relationship between food insecurity (FI) and depression, but had no significant influence on FI's associations with anxiety or stress. In practical terms, FI had a substantially more severe depressive impact on lower-income individuals (i.e., those near the poverty threshold). In contrast, FI-related anxiety and stress levels did not differ markedly across income strata. These findings suggest that limited financial resources exacerbate the risk of depression under conditions of FI. In contrast, FI-related anxiety and stress appear to affect food-insecure individuals more uniformly, irrespective of income. This selective influence of income on depression challenges the assumption that fundamental socioeconomic or demographic indicators uniformly determine the mental health impacts of FI. Instead, these results point to deeper social and structural determinants such as cultural displacement, social integration, and systemic inequalities as pivotal factors in shaping mental health outcomes among food-insecure migrants and refugees [5,21].

## Strengths and limitations

Our review has notable strengths, including adherence to gold standard methods such as registering the study protocol in advance of conducting the review on PROSPERO, employing a rigorous multi-stranded search strategy to supplement our database searches with forward and backward citation chaining, rigorous methodologies, comprehensive subgroup analyses, and extensive comparative analyses of existing literature. Nonetheless, certain limitations are acknowledged. The included studies were cross-sectional, which restricted causal inference. Observed heterogeneity attributable to methodological differences, varied FI measurement tools, and diverse socio-cultural contexts necessitated cautious interpretation. Additionally, the detected publication bias may indicate an overrepresentation of small studies reporting significant results.

## Conclusion

In conclusion, this systematic review and meta-analysis found consistent associations between food insecurity and poorer mental health outcomes, depression, anxiety, and stress, among migrants and refugees living in high-income countries. Given that most evidence is observational (cross-sectional studies), these findings should be interpreted with caution; nevertheless, they underline the value of screening for food insecurity and linking affected individuals to culturally responsive food and mental health support. Future longitudinal and intervention studies are needed to clarify pathways and to test individual, community, and system-level responses.

## Supporting information

**S1 File. Published protocol: Food insecurity and psychological stress among migrants and refugees in high-income countries: Protocol for a systematic review and meta-analysis.**
(DOCX)

**S1 Checklist. PRISMA 2020 checklist for the manuscript.**
(DOCX)

**S2 Checklist. PRISMA 2020 for Abstracts checklist.**
(DOCX)

**S1 Appendix. Full database search strategies (Medline, Scopus, Web of Science, PsycINFO, and Embase).**
(DOCX)

**S1 Table. Study-level methodological appraisal (Joanna Briggs Institute checklist) for included studies.**
(DOCX)

**S1 Fig. Funnel plot for the association between food insecurity and depression.**
(DOCX)

**S2 Fig. Funnel plot for the association between food insecurity and anxiety.**
(DOCX)

## Acknowledgments

We acknowledge the support of Western Sydney University's library services in obtaining full-text articles and providing technical support for the review software. Finally, we are grateful to the peer reviewers of our protocol, whose feedback helped improve the quality of this review.

## Author contributions

**Conceptualization:** Resom Berhe, Amit Arora, Kingsley E. Agho.

**Data curation:** Resom Berhe, Amit Arora, Kanchana Ekanayake, Kingsley E. Agho.

**Formal analysis:** Resom Berhe, Amit Arora, Kingsley E. Agho.

**Investigation:** Resom Berhe.

**Methodology:** Amit Arora, Kingsley E. Agho.

**Supervision:** Amit Arora, Kingsley E. Agho.

**Visualization:** Resom Berhe, Kingsley E. Agho.

**Writing – original draft:** Resom Berhe.

**Writing – review & editing:** Resom Berhe, Amit Arora, Kingsley E. Agho.

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
