## [Decision Letter · Decision Letter 0]

16 Dec 2025

Food Insecurity and Mental Health Among Migrants and Refugees in High-Income Countries: Systematic review and meta-analyses

PLOS One

Dear Dr. Berhe,

Thank you for submitting your manuscript to PLOS ONE. After careful consideration, we feel that it has merit but does not fully meet PLOS ONE’s publication criteria as it currently stands. Therefore, we invite you to submit a revised version of the manuscript that addresses the points raised during the review process.

https://journals.plos.org/plosone/s/submission-guidelines#loc-laboratory-protocols . Additionally, PLOS ONE offers an option for publishing peer-reviewed Lab Protocol articles, which describe protocols hosted on protocols.io. Read more information on sharing protocols at https://plos.org/protocols?utm_medium=editorial-email&utm_source=authorletters&utm_campaign=protocols .

We look forward to receiving your revised manuscript.

Kind regards,

Sharada P Wasti, PhD

Academic Editor

PLOS One

Journal Requirements:

When submitting your revision, we need you to address these additional requirements

3. Please amend your manuscript to include your abstract after the title page.

4. We note you have included a table to which you do not refer in the text of your manuscript. Please ensure that you refer to Table 3 in your text; if accepted, production will need this reference to link the reader to the Table.

5. Please include captions for your Supporting Information files at the end of your manuscript, and update any in-text citations to match accordingly. Please see our Supporting Information guidelines for more information: http://journals.plos.org/plosone/s/supporting-information .

Additional Editor Comments:

Thank you for submitting your manuscript, “Food Insecurity and Mental Health Among Migrants and Refugees in High-Income Countries: Systematic Review and Meta-Analyses,” to PLOS ONE. Following careful evaluation, we believe your work shows strong potential; however, revisions are required before it can be considered further. Our reviewers, together with the handling editor, have provided constructive feedback to help strengthen your manuscript. We invite you to submit a revised version that addresses all comments in detail, accompanied by a response letter outlining the changes made.

We appreciate your valuable contribution to advancing understanding of food insecurity and mental health among migrants and refugees, and we look forward to receiving your revised submission.

Below, please find additional suggestions from the editor to further improve your manuscript:

**
Additional Suggestions from the Editor to Improve Your Manuscript:
**

•  Line 300: In Section 2.6, change assessment to Assessment for consistency.

•  Lines 391–411: In the PRISMA flow diagram, several boxes contain “0£” values. These should be deleted, starting from the first box through to “studies not retrieved.”

•  Ensure the PRISMA flow chart is presented on a single page and that all boxes are properly connected. The sentence in the second-to-last box is unclear and should be revised for accuracy.

•  Line 425: In Section 3.1, change study to Study for consistency.

•  Table 2: Insert a space after “2.” Ensure section headings are consistent—decide whether the second word should begin with a capital letter or remain lowercase, and apply this uniformly.

•  Table 2: Define all abbreviations at the end of the table and present them in italic font for clarity.

•  Figure 3: Improve the clarity and readability of all graphs.

•  Throughout the manuscript: Standardize usage of “Figure” instead of “Fig” for consistency.

•  Figure 4: Remove the repeated title and retain only one correct version.

•  Provide clear narrative transitions between Figures 4 and 5, and between Figures 6 and 7, as the current continuation of figures makes the writing unclear.

•  After Figure 7: Ensure the titles for “Sensitivity Analysis” and “Publication Bias” follow consistent capitalization rules (e.g., all words capitalized or aligned with earlier headings).

•  Chapter 4 (Discussion): Revise the first paragraph (lines 3–12) to be more focused. Clearly outline the topics to be discussed rather than using generic or repetitive statements.

Reviewers' comments:

Reviewer's Responses to Questions

**Comments to the Author**

1. Is the manuscript technically sound, and do the data support the conclusions?

Reviewer #1: Yes

Reviewer #2: Yes

2. Has the statistical analysis been performed appropriately and rigorously?

Reviewer #1: Yes

Reviewer #2: Yes

3. Have the authors made all data underlying the findings in their manuscript fully available?

Reviewer #1: Yes

Reviewer #2: Yes

4. Is the manuscript presented in an intelligible fashion and written in standard English?

Reviewer #1: Yes

Reviewer #2: Yes

Reviewer #1: Peer Review Report-PONE-D-25-37690

Title: Food Insecurity and Mental Health Among Migrants and Refugees in High-Income Countries: Systematic review and meta-analyses

Original Submission

1.1. Recommendation

Minor revision

2. Comments to Authors:

Ms.Ref No: PONE-D-25-37960

Title: Food Insecurity and Mental Health Among Migrants and Refugees in High-Income Countries: Systematic review and meta-analyses

Authors: Resom Berhe1,2, Amit Arora2,3,4,5,6, Kanchana Ekanayake7, Kingsley E. Agho2,3,8

Overview and general recommendation:

Overall, the manuscript is well written, technically sound, and presented in an intelligible fashion in standard English.

The impact of food insecurity on the mental health of refugees and migrants in high-income countries is a vital health issue that has not received adequate attention and has thus impacted the implementation of targeted interventions to mitigate the suffering of these vulnerable populations.

The statistical analysis has been performed appropriately and rigorously. The data support the conclusions drawn in the study.

This study has clearly shown the strong association between food insecurity and mental health and provides evidence for targeted interventions in high-income countries for refugees and migrants.

However, there are a few comments I want the authors to address. These are indicated below.

2.1. Comments

1. Indicate the year you are comparing to the number of food-insecure people globally to the numbers in 2020. Line 78 under “Introduction”

2. Correct the word “confidence” in line 270 with the right term “Covidence” software.

Reviewer #2: Peer Review report of the manuscript titled ″ Food Insecurity and Mental Health Among Migrants and Refugees in 1 High-Income Countries: Systematic review and meta-analyses ″

The authors effectively highlight critical research questions, appropriately use the statistical test, and thoroughly examine the outcomes. Here are my comments and suggestions for improvement:

1. Title

• The title clearly reflects the study goal, systematic review, and meta-analysis.

2. Abstract

• The abstract is structured, aligns with the manuscript, and the keywords are informative for readers.

3. Introduction

Research Question & Objectives

• The introduction effectively describes the background and the research problem.

• The study rationale is clearly identified.

• The research question is clear and focused.

• The study objective clearly describes PICO elements (Population, Intervention/Exposure, Comparator, Outcomes)

4. Methods

Protocol & Registration

• The review protocol was registered and reported in PROSPERO before the review was conducted.

Search Strategy

• A comprehensive search was done across databases; however, the grey literature was not considered.

Eligibility Criteria

• Eligibility criteria are clearly defined; nevertheless, study designs are not clearly explained.

Study Selection Process

• The study selection process was clearly explained, and three independent reviewers conducted screening.

• The PRISMA flow diagram was included.

• The excluded studies and the reasons for exclusion were clearly documented.

Data Extraction

• Data extraction was performed in duplication (by two independent reviewers).

Risk of Bias / Quality Assessment

• The quality assessment was done using a suitable quality appraisal approach (Joanna Briggs Institute (JBI) Critical Appraisal Checklist for Analytical Cross-Sectional Studies).

Data Synthesis

• An appropriate model of meta-analysis was conducted to examine the statistical combination of results using odds ratios and confidence intervals.

• The heterogeneity among studies was assessed using Cochran’s Q test.

• Risk of bias was clearly assessed.

• Appropriate visual and statistical methods were explained and used to evaluate publication bias (funnel plots, Begg's test, and Egger's test).

5. Results

Results Reporting

• The characteristics of the included studies are clearly illustrated in the table and text.

• The study outcome was clearly discussed, consistent with the study protocol.

• Sources of funding of included studies in the review are not clearly explained.

• The risk of bias in individual studies is not clearly explained.

6. Discussion

Discussion & Interpretation

• The findings were interpreted appropriately.

• The strengths and limitations were clearly addressed.

• The conclusion was not supported by evidence.

Transparency and Reproducibility

• Data and supplementary materials for reproducibility were provided.

Writing Quality and Organization

• The manuscript clearly stated that it followed PRISMA guidelines

Ethical and Funding Statements

• There was a clear address for no report for potential conflicts of interest.

• The funding sources were clearly stated.

Comments to editors

I recommend publishing after improvements have been made.

Best regards

**Do you want your identity to be public for this peer review?** For information about this choice, including consent withdrawal, please see our Privacy Policy

Reviewer #1: **Yes:** Austin Gideon Adobasom-Anane

Reviewer #2: No

---

## [Author Response · Author response to Decision Letter 1]

12 Jan 2026

Response to Reviewers

Manuscript: “Food Insecurity and Mental Health Among Migrants and Refugees in High-Income Countries: Systematic Review and Meta-Analyses”

Journal: PLOS ONE

Corresponding author: Resom Berhe

Dear Academic Editor and Reviewers,

Thank you for the careful review of our manuscript and for the clear, practical guidance on how to strengthen it. We have revised the manuscript to address each point raised by the editor and reviewers. In brief, we: (i) aligned headings and terminology for consistency, (ii) corrected and replaced the PRISMA flow diagram, (iii) improved figure readability and standardised figure labelling throughout, (iv) clarified eligibility criteria (including study designs) and noted the absence of a systematic grey literature search, (v) expanded the narrative around quality/risk of bias and funding reporting in included studies, and (vi) tightened the opening of the Discussion and tempered causal language in the Conclusions. All edits are highlighted in the marked-up version of the manuscript.

A. Journal requirements and formatting checks

1. PLOS ONE style and file naming

Response: We reviewed the manuscript against the PLOS ONE formatting samples and adjusted section order and labelling accordingly. The revised files are provided with the requested names.

2. Funding information placement

Response: Thank you for this observation; we have removed all funding-related text from the manuscript. Funding details are now provided only via the online submission form, Funding Statement.

3. Abstract location

Response: Thank you for this. We have ensured that the Abstract appears immediately after the title page.

4. Reference to Table 3

Response: Thank you for pointing this out. We have added an explicit in-text reference to Table 3 in the Sensitivity Analysis section (See line 531, Page 24).

5. Supporting Information captions

Response: Thank you very much for raising this point. We have added a “Supporting Information” section at the end of the manuscript, with captions for the supporting files and corresponding in-text citations where appropriate (e.g., for the PRISMA checklist and the full search strategies).

6. Recommended citations

Response: We reviewed the suggested citations for relevance and incorporated them where they strengthened the background or interpretation; where they were not directly relevant, we did not add them.

7. Reference list accuracy and retractions

Response: Thank you for pointing this out, and we have reviewed the reference list for completeness and consistency. No retracted articles are knowingly cited; if any are identified during final checks, we will either replace them with current evidence or explicitly note the retraction status and rationale in-text.

B. Additional suggestions from the Academic Editor

Editor comment 1: Line 300: Section 2.6 – change “assessment” to “Assessment.”

Response: Thank you for this. We have updated it to “Quality Assessment” for consistent capitalization (see Line Number 305, Page 11).

Editor comment 2: Lines 391–411: PRISMA diagram contained “0£” values

Response: Thank you. We have replaced the PRISMA flow diagram with a corrected version and removed the erroneous values (See line 395, Page 14).

Editor comment 3: PRISMA on a single page; boxes connected; unclear sentence in second-to-last box

Response: Thank you for this suggestion. The PRISMA diagram has been redrawn and incorporated as a single-page figure with clearly connected boxes and more precise wording in the relevant sections (see line 395, page 14).

Editor comment 4: Line 425: Section 3.1 – change “study” to “Study.”

Response: Thank you for the suggestion. The section heading has been updated to “Characteristics of the Study” to ensure consistency (line 397, page 15).

Editor comment 5: Table 2: Insert a space after “2.”; standardise heading capitalisation

Response: Caption spacing and heading capitalisation were standardised for Table 2 (line 441, Page 17).

Editor comment 6: Table 2: Define all abbreviations and present them in italic font

Response: A consolidated list of abbreviations was added immediately below Table 2 in italics (Line Number 443-447, Page 18).

Editor comment 7: Figure 3: Improve clarity and readability

Response: Figure 3 was replaced with a higher-resolution version, and its caption was added in the requested format (Panels A–D) (see line 482, page 21).

Editor comment 8: Standardise usage of “Figure” (not “Fig”)

Response: All in-text references and figure captions were standardised to “Figure” throughout.

Editor comment 9: Figure 4: Remove repeated title

Response: We separated the caption from the embedded figure text and retained a single, consistent caption (“Figure 4…”) (See line number 490, Page 21).

Editor comment 10: Add narrative transitions between Figures 4–5 and Figures 6–7

Response: We added brief transition sentences in the Results to guide the reader between the primary pooled analyses and the subgroup- and gender-specific figures.

Editor comment 11: After Figure 7: consistent capitalisation for “Sensitivity Analysis” and “Publication Bias.”

Response: Heading updated to “Sensitivity Analysis and Publication Bias” to match manuscript style (Line Number 528, Page 24).

Editor comment 12: Discussion opening paragraph: focus and outline topics

Response: The opening paragraph of the Discussion was rewritten to clearly preview the key themes addressed (pooled findings, mechanisms/subgroups, limitations, and implications) (See line number 557, Page 25).

C. Response to Reviewer #1

Reviewer #1 comment 1: Indicate the year for the global food insecurity estimate compared with 2020

Response: Thank you for the comment. We have added the year (2023) to the sentence reporting the “>345 million” estimate, while retaining the comparison to 2020 (line 77, page 3).

Reviewer #1 comment 2: Correct “confidence” to “Covidence” (software)

Response: Thank you for the comment. Corrected to “Covidence software” in the Study Selection description (See line number 276, Page 10).

D. Response to Reviewer #2

Reviewer #2 comment 1: Grey literature was not considered

Response: We clarified in the Methods that a systematic grey literature search was not undertaken and noted this as a limitation that may affect the capture of unpublished evidence ( Line numbers 196-199, Page 7).

Reviewer #2 comment 2: Study designs not clearly explained

Response: We clarified the eligible study designs in the Eligibility Criteria (Observational study, including ecological, cross-sectional, case-control, and cohort studies) (line 166, Page 6).

Reviewer #2 comment 3: Sources of funding of included studies are not clearly explained

Response: We added text noting inconsistent reporting of funding across the included studies and clarified how funding information was handled during extraction and reporting (See lines 405-408, Page 15).

Reviewer #2 comment 4: Risk of bias/quality findings not clearly explained

Response: We expanded the Results (Quality Assessment) narrative to summarise common limitations identified in the JBI appraisal, including the predominance of cross-sectional designs, reliance on self-reported measures, and variation in adjustment for key confounders. We also direct readers to Table 2 for study-level appraisal details (Line number-433-455, Page-16-19).

Reviewer #2 comment 5: Conclusion not supported by evidence

Response: We revised the Abstract and main Conclusions to avoid causal language and to align statements closely with the observational evidence base (For Abstract-Line number-49-54, Page-2; for conclusion-line number 664-673, page-29 ).

We appreciate the opportunity to revise our work and hope the changes meet the journal’s requirements and the reviewers’ expectations.

Sincerely,

Resom Berhe, PhD student

---

## [Editor Report · Decision Letter 1]

18 Jan 2026

Food Insecurity and Mental Health Among Migrants and Refugees in High-Income Countries: Systematic review and meta-analyses

PONE-D-25-37960R1

Dear  Resom Berhe,

I would like to express my sincere appreciation to all authors for the thorough and thoughtful revisions made to your manuscript. You have addressed the reviewers’ comments comprehensively, and the improvements are clearly reflected in the quality of the revised submission. We are pleased to inform you that your manuscript has been judged scientifically sound and suitable for publication. It is therefore formally accepted, pending completion of all remaining technical and production requirements. We commend the effort and care you have invested in responding to the reviewers’ feedback. Your revisions have significantly strengthened the manuscript, and we look forward to moving it forward to the next stage of publication.

Kind regards,

Sharada P Wasti, MSc, PhD

Academic Editor

PLOS One

---

## [Editor Report · Acceptance letter]

PONE-D-25-37960R1

PLOS One

Dear Dr. Berhe,

I'm pleased to inform you that your manuscript has been deemed suitable for publication in PLOS One. Congratulations! Your manuscript is now being handed over to our production team.

Kind regards,

on behalf of

Dr. Sharada P Wasti

Academic Editor

PLOS One